# Extracellular Calcium Ion Concentration Regulates Chondrocyte Elastic Modulus and Adhesion Behavior

**DOI:** 10.3390/ijms221810034

**Published:** 2021-09-17

**Authors:** Xingyu Shen, Liqiu Hu, Zhen Li, Liyun Wang, Xiangchao Pang, Chun-Yi Wen, Bin Tang

**Affiliations:** 1Department of Biomedical Engineering, Southern University of Science and Technology, Shenzhen 518055, China; 11749079@mail.sustc.edu.cn (X.S.); 11930748@mail.sustech.edu.cn (L.H.); 2School of Chemistry and Environmental Engineering, Jiangsu University of Technology, Changzhou 213001, China; lizhen198458@163.com; 3Center for Biomechanical Research, Department of Mechanical Engineering, University of Delaware, Newark, DE 19716, USA; lywang@udel.edu; 4College of Materials Science and Engineering, Central South University of Forestry and Technology, Changsha 410004, China; T20172373@csuft.edu.cn; 5Department of Biomedical Engineering, The Hong Kong Polytechnic University, Hong Kong 999077, China; chunyi.wen@polyu.edu.hk

**Keywords:** extracellular calcium ion, elastic modulus, adhesion, migration

## Abstract

Extracellular calcium ion concentration levels increase in human osteoarthritic (OA) joints and contribute to OA pathogenesis. Given the fact that OA is a mechanical problem, the effect of the extracellular calcium level ([Ca^2+^]) on the mechanical behavior of primary human OA chondrocytes remains to be elucidated. Here, we measured the elastic modulus and cell–ECM adhesion forces of human primary chondrocytes with atomic force microscopy (AFM) at different extracellular calcium ion concentration ([Ca^2+^]) levels. With the [Ca^2+^] level increasing from the normal baseline level, the elastic modulus of chondrocytes showed a trend of an increase and a subsequent decrease at the level of [Ca^2+^], reaching 2.75 mM. The maximum increment of the elastic modulus of chondrocytes is a 37% increase at the peak point. The maximum unbinding force of cell-ECM adhesion increased by up to 72% at the peak point relative to the baseline level. qPCR and immunofluorescence also indicated that dose-dependent changes in the expression of *myosin* and *integrin β1* due to the elevated [Ca^2+^] may be responsible for the variations in cell stiffness and cell-ECM adhesion. Scratch assay showed that the chondrocyte migration ability was modulated by cell stiffness and cell-ECM adhesion: as chondrocyte’s elastic modulus and cell-ECM adhesion force increased, the migration speed of chondrocytes decreased. Taken together, our results showed that [Ca^2+^] could regulate chondrocytes stiffness and cell-ECM adhesion, and consequently, influence cell migration, which is critical in cartilage repair.

## 1. Introduction

Osteoarthritis (OA) is a common joint disorder and a leading cause of physical disability; one of its characteristics is articular cartilage degeneration [1]. There is no effective way to cure OA at present, despite its negative impacts on patients’ life quality. There are many risk factors of OA, including but not limited to obesity, aging, genetics, and trauma to the joint [2,3]. Abnormal extracellular calcium ion concentration was reported in osteoarthritic cartilage and in severe OA cases where calcium crystals were found in the joint fluid and hyaline articular cartilage of patients [4,5,6]. Therefore, extracellular calcium ([Ca^2+^]) may play an important role in OA pathogenesis [7]. Furthermore, in the synovial fluid, the [Ca^2+^] of OA patients was significantly higher than that of the healthy people [8,9].

Chondrocytes, the sole cell type in cartilage, are responsible for maintaining the integrity of the cartilage matrix and secreting matrix-associated proteins [10]. Although the concept is still being debated, the movement of chondrocytes in vivo can be inferred from various findings based on in vitro cell cultures, ex vivo organ cultures and 3D human cartilage, and indications of mobile chondrocytes include the formation and extension of chondrocyte cell processes in the vicinity of cartilage damage and the outgrowth of chondrocytes at the wound margin [10,11,12,13]. Since chondrocytes in vivo are surrounded by a proteoglycan-rich pericellular matrix and an extensive collagen network, the migration of chondrocytes is a challenging process involving the deformation of the cell body and coordination of the binding and unbinding of the cell–extracellular matrix (ECM) adhesion. Therefore, in this migration process, the mechanical properties of chondrocytes, such as elastic modulus and adhesion, should play an important role.

It is known that cells with a higher migration ability are often softer than their counterparts, due to their high cytoskeleton deformability [14,15,16]. Actin filaments are major components of the cytoskeleton that contribute to cell stiffness and are involved in cell deformation and migration [17]. Myosins are also important components of the cytoskeleton and bind with actin filaments to generate cell contractile forces. Previous studies have demonstrated that calcium ions have an effect on actin and myosin in certain cell types [18]. For instance, calcium ions control actin organization [19,20], influence actin–myosin interaction [21], and change myosin conformation [22]. These studies provide us with indications that calcium ions may affect cell stiffness. On the other hand, cells need to be released from their existing adhesion to ECM at the trailing edge and form new adhesions at the leading edge during cell movement [23]. Cell–ECM adhesions are mediated by integrin (mainly integrin β1) in chondrocytes [24,25]. Previous reports indicated that calcium ions can regulate integrin-mediated adhesion and cell migration. For example, extracellular calcium ions regulate osteoclast’s integrin expression and change the adhesion, then influence the cell’s migration [26]. These results suggest that extracellular calcium ions may also regulate chondrocyte adhesion. However, the effects of increased calcium levels on the mechanical properties and function of chondrocytes are not clear.

Therefore, it is reasonable to hypothesize that extracellular calcium ions regulate chondrocyte’s stiffness and cell–ECM adhesion and further influence cell migration, which may be related to the tissue repair abilities. To test this hypothesis, we measured the elastic modulus of chondrocytes and the cell–ECM adhesion force at a different [Ca^2+^] using human primary chondrocytes with atomic force microscopy (AFM). We also quantified the migration abilities of chondrocytes and analyzed the cytoskeleton structure. Moreover, several cytoskeleton-related genes and protein expressions were investigated to explore the possible reasons for the observed changes in cell mechanical properties with a different [Ca^2+^].

## 2. Results

### 2.1. Chondrocyte Stiffness

The Young’s modulus of human chondrocytes showed an increasing and decreasing trend as calcium ion concentration increased, by up to 36.8% at the peak point. A typical indentation curve, including the approaching segment, holding segment, and retracting segment, for the elastic modulus of chondrocytes cells, was shown in Figure 1A. The Young’s modulus of the cell was obtained from the holding segment and retracting segment of the indentation curves. The Young’s modulus of human chondrocytes increased by 26% from the case of 1.75 mM (1202 ± 250 Pa, *p* > 0.05) to 2.25 mM (1521 ± 75 Pa) and continued to increase by 37% at 2.75 mM (1644 ± 396 Pa, *p* < 0.0001). Then, the Young’s modulus decreased at 3.25 mM (1402 ± 211 Pa, *p* < 0.001) and 3.75 mM (1510 ± 155 Pa, *p* < 0.001), relative to 2.75 mM (Figure 1B). The precise values were shown in Appendix A.

### 2.2. Chondrocyte Adhesion to ECM

The maximum unbinding force (*F_max_*) and work of detachment (*W_d_*) increased by up to 48% and 72% as the calcium ion concentration increased. Both *F_max_* and *W_d_* had tendencies to first increase and then decrease. A measurement scheme of the adhesion between chondrocytes and extracellular matrix on AFM tip is shown in Figure 2A. *F_max_* and *W_d_* could be obtained from the approaching segment of the indentation curves (Figure 2B). *F_max_* in the control group was 130 ± 90 pN. *F_max_* increased by up to 48% when [Ca^2+^] increases from 1.75 mM to 2.75 mM (1103 ± 439 pN vs. 1632 ± 724 pN, *n* = 90, *p* = 0.0001, Figure 2C). Compared with that of 1.75 mM, the *F_max_* of the other three groups (2.25 mM, 3.25 mM, and 3.75 mM) showed little increase, but there is no significant difference among them (*p* > 0.05).

*W_d_* in the control group was 1.2 ± 1.1 × 10^−16^ J. *W_d_* increased by 31% from 1.75 mM to 2.25 mM (1.3 ± 0.8 × 10^−15^ J vs. 1.8 ± 0.6 × 10^−15^ J), and increased by 72% at 2.75 mM (1.3 ± 0.8 × 10^−15^ J vs. 2.3 ± 0.9 × 10^−15^ J, *p* < 0.0001). Although *W_d_* shows a slight increase for [Ca^2+^], between 3.75 mM and 1.75 mM; there is no significant difference between them (*p* > 0.05, Figure 2D). The precise values are shown in Appendix A.

### 2.3. Chondrocyte Migration

To assay the cell migration ability of chondrocytes, the scratch assay was carried out under a different [Ca^2+^]. Representative scratch assay images for 1.75 mM are shown in Figure 3A at 0 h and 48 h. The migration distance varied with [Ca^2+^] in a biphasic manner (Figure 3B). Migration distance decreased by 21% (*p* = 0.02) between 2.75 mM (402 ± 60 μm) and 1.75 mM (509 ± 17 μm). Compared to the normal calcium level, the cell migration ability decreased with the increasing [Ca^2+^]. The MTT assay was utilized to measure the proliferation of chondrocytes at days 1 and 3. The results showed no difference between each group at day 3, exempting the influence of cell proliferation on the variant migration distance (Appendix A).

### 2.4. Chondrocyte Cytoskeleton and Adhesion-Related Proteins Expression

F-actin was detected to analyze the variance in the chondrocyte elastic modulus. A typical F-actin fluorescence image at 1.75 mM [Ca^2+^] was shown in Figure 4A. The fluorescence density of F-actin filaments had no significant difference among the different groups (n = 60 cells/group, *p* > 0.05, Figure 4B).

Immunofluorescence staining was used to confirm the expression of myosin and integrin β1 in different groups. The fluorescence images of integrin β1 and myosin at 1.75 mM [Ca^2+^] are shown in Figure 6A,C, respectively. Compared with the 1.75 mM group, the fluorescence density of integrin β1 was significantly increased by 20% in 2.75 mM group (24.6 ± 7.9 vs. 29.5 ± 5.8, n = 60, *p* = 0.0008). With the increase in [Ca^2+^], the fluorescence density decreased by 10%, from 2.75 mM to 3.75 mM (29.5 ± 5.8 vs. 26.6 ± 5.4, n = 60, *p* = 0.015, Figure 6B). A similar trend was also observed in myosin (Figure 6D). The fluorescence density was significantly increased (10.9 ± 2.1 vs. 15.4 ± 1.7 and 15.5 ± 3.2, n = 60, *p* < 0.0001) by 50% in the lower [Ca^2+^] groups (2.25 mM and 2.75 mM), and then significantly decreased by 20% (15.5 ± 3.2 vs. 12.6 ± 2.4 and 12.1 ± 1.9, n = 60, *p* < 0.0001) with the higher [Ca^2+^] (3.25 mM and 3.75 mM, Figure 6D). The similar expression trends of *myosin* and *integrin* were also shown in the Western blot results (Appendix A).

### 2.5. Chondrocyte Adhesion to ECM after Integrin Inhibition

*F_max_* and *W_d_* decreased after treatment with 10 nM CWHM-12 compared with the untreated groups. The value of *F_max_* in five groups was between 300 ± 200 pN (2.25 mM) and 410 ± 217 pN (2.75 mM) (*n* = 60 cells/group, Figure 7A). The value of *W_d_* in five groups was between 3.6 ± 2.8 × 10^−16^ J (1.75 mM) and 4.6 ± 3.1 × 10^−16^ J (3.75 mM) (*n* = 60 cells/group, Figure 7B). Compared with the untreated groups, the *F_max_* decreased between 77% (3.75 mM, *p* < 0.0001) to 69% (2.25 mM, *p* < 0.0001) after CWHM-12 treatment (Appendix A). The *W_d_* decreased between 71% (3.75 mM, *p* < 0.0001) and 83% (2.75 mM, *p* < 0.0001) compared with the five untreated groups (Appendix A).

## 3. Discussion

In recent years, the high concentration of calcium in the joint fluid of OA patients has emerged as an important hallmark in the pathogenesis of OA, and emerged as a critical simulation in cells. Despite its physiological importance, the relationship between a high [Ca^2+^] level and chondrocyte mechanics and chondrocyte migration is still ambiguous. In this investigation, we measured chondrocyte stiffness and chondrocyte–ECM adhesion and explained their relationship with cell migration. Furthermore, we demonstrated the critical role of myosin and integrin in chondrocyte mechanics and migration. Our work leads to an important advance in the understanding of the role of [Ca^2+^] in OA pathogenesis and also helps to reveal the correlation between cell mechanics and cell migration

Cell mechanical properties are closely related to cellular activities, such as migration, differentiation, and metastasis [14]. The nanomechanical characterization of chondrocytes at different extracellular calcium concentrations helps in the observation of the abnormal mechanical properties of chondrocytes between OA and normal cartilage. AFM was employed to measure the mechanical properties of chondrocytes in this study. The indentation depth of the AFM tip was approximately 0.5 μm. At this indentation depth, the elastic modulus is mainly determined by the cortical cytoskeleton, which is comprised of actin filaments and myosin, with a thickness of approximately 50–200 nm [14,27,28]. Cortical actin filaments interact with myosin, which is responsible for maintaining the contractile force. When the [Ca^2+^] level changes, the cell can detect small fluctuations in the calcium level and respond by regulating actin and myosin [29]. Actin and myosin can be regulated by calcium in several different ways. Firstly, increasing [Ca^2+^] induces an influx of calcium ions and then causes a cascade of calcium signals [30]. Intracellular calcium ions regulate actin arrangement, altering the mechanical properties and migration of cells [31,32]. Calcium plays a role in the phosphorylation of myosin and regulates myosin–actin interactions in non-muscle mammalian cells [22,33]. Secondly, changes in [Ca^2+^] will cause variations in the expression of skeletal-related genes, mediated by calcium-sensitive receptors, leading to the re-arrangement of F-actin [34,35]. In the present study, increasing the [Ca^2+^] level induces the expression of *myosin* to increase from 1.75 mM to 2.75 mM, and decrease from 2.75 mM to 3.75 mM, which can be responsible for the alternation in the chondrocyte stiffness. Santos-Argumedo et al. also reported that the decrease in cell elastic modulus after myosin was due to knockout [36]. The immunofluorescence and Western blot experiments confirmed that the expression of *myosin* is nearly in accordance with its gene expression results. 

Cell–ECM adhesion forces were measured by an AFM tip coated with the chondrocyte extracellular matrix. Cell–ECM adhesion is mainly mediated by transmembrane protein integrin, which contains 24 heterodimers, formed of β subunits and α subunits. As chondrocyte ECM mainly consists of collagen Ⅱ and aggrecan, the adhesion is principally mediated by integrins β1 subunits, such as α1β1, α2β1 and α10β1 in chondrocytes [25]. In the present study, adhesion forces and detachment work varied with [Ca^2+^], and the patterns were consistent with the expression of integrin β1. A higher expression level of integrin suggests that there may be more molecular binding to ECM, producing stronger adhesion forces that require more energy to break during detachment. Our results demonstrate that cell–ECM adhesion can be regulated by the extracellular calcium concentration, through integrinβ1. Calcium ions may regulate integrin in the following ways. Firstly, integrin-mediated adhesion can be regulated by calcium signals from inside and outside the cells. Integrin mediates calcium signaling from the extracellular domain (‘outside-in’ signaling), and then activates intracellular signal pathways. Intracellular feedback signals subsequently regulate integrin-mediated adhesion (‘inside-out’ signaling) [37,38]. Secondly, the regulation of Ca^2+^ to integrin presents a bipolar trend and Ca^2+^ plays an important role in stabilizing the integrin structure to mediate integrin-ligand binding. There are three or four Ca^2+^ binding sites in integrin, which regulate integrin–ligand binding affinity. Calcium ions bind to integrin and change the conformation, thus influencing the integrin–ligand binding affinity [39]. The previous ligand binding assay demonstrated that when calcium ion is in a lower concentration, Ca^2+^ binds with ligand-binding sites such as ligand-associated metal-binding site (LIMBS) and metal-ion dependent adhesion site (MIDAS), and plays a positive role in integrin–ligand adhesion. When calcium ion increases to a higher concentration level (1–10 mM), Ca^2+^ binds with a ligand-bind site such as ADMIDAS (adjacent to MIDAS), and plays a negative role in integrin–ligand adhesion [40,41]. In this study, as Ca^2+^ remains at a high concentration, Ca^2+^ binds with ADMIDAS, causing an integrin to remain at a low level of affinity and present a smaller detachment force. Taken together, the expression of integrin and integrin affinity changes, causing cell–ECM adhesion, presents a bipolar trend in a calcium concentration ranging from 2.75 mM to 3.75 mM. To further verify the integrin’s impact on the mediation of cell–ECM adhesion, CWHM-12 was used to inhibit integrin [42]. The sharp drop in the adhesion forces and *W_d_* after CWHM-12 treatment demonstrated that integrin is the main molecular in the mediation of chondrocyte–ECM adhesion.

Cell migration is a complex process that underlies tissue formation, maintenance and regeneration. Each cell type employs a particular migration manner in a given environment [43]. There are numerous factors that can affect cell migration, such as ECM architecture, cell–cell interactions, the property of the individual cell, etc. ECM architecture, such as 2D and 3D culture environments, can also influence the cell migration process. The migration of chondrocytes cultured in a 3D environment normally has the following steps: edge protrusion, integrin-mediated focal interactions to the substrate, actomyosin-mediated cell contraction and rear-end retraction [44]. In a 2D environment, chondrocyte migration requires unilateral adhesion to the substrate and the formation of a transient but stable-enough adhesion, led by a leading lamellipodia [45,46]. Both 2D and 3D migration manners involve cell deformation and cell detachment. Usually, cells with lower stiffness are easier to deform during migration, and higher cell motility usually corresponds with a lower stiffness [14]. High integrin expression levels are correlated with high attachment forces, and high detachment forces and a relatively slower dissociation rate at the trailing edge of the cell [47,48]. In the present study, when the [Ca^2+^] increased from 1.75 mM to 2.75 mM, both the elastic modulus and the adhesion forces of chondrocytes increased, resulting in a decreased migration ability in chondrocytes. When [Ca^2+^] increased from 2.75 mM to 3.75 mM, the opposite effects were seen in the chondrocytes’ elastic modulus, adhesion forces, and cell migration ability.

There are several limitations to the present study. As mentioned above, the migration pattern of cells in the 2D environment is different from that in the 3D environment. In the present study, chondrocytes were cultured in a 2D petri dish, so the migration was not fully consistent with the physiological environment. Some studies have shown that chondrocytes have relatively weak migration abilities. The apparent cell migration in vivo predominantly occurred in the collagen fibril orientation, and occurred at a slower rate than other mammalian cells in 2D culture [10]. The primary culture of chondrocytes isolated from human cartilage has been a useful model for studies of related diseases. However, chondrocytes cultured in monolayer tended towards de-differentiation and the cartilage-specific phenotype could not be maintained with the passage of time. Mature articular chondrocytes expressed type Ⅱ collagen and aggrecan as markers and de-differentiated chondrocytes presented a fibroblastic phenotype. In the present study, the expression results of *collagen I* and *collagen II* indicated that chondrocytes were slightly de-differentiated at the fourth day, but the effect was not severe. When cultured for 12 days, the ratio of *collagen I* to *collagen II* was in the dozens, which indicated that the chondrocytes’ de-differentiation became severe. The expression levels of *collagen II* and aggrecan in each group were almost the same, which indicated that [Ca^2+^] does not significantly affect chondrocytes’ differentiation (Appendix A).

Taken together, the mechanical properties of chondrocytes are closely related to various important biological behaviors, such as migration and proliferation. Multitudinous stimuli factors from intracellular and extracellular domains can influence the chondrocyte’s mechanical properties and regulate its biological behavior. Calcium’s involvement in intracellular signal transduction in cells has been demonstrated. In this study, we measured the elastic modulus and cell–ECM adhesion force of chondrocytes cultured in vitro using AFM. The measured results suggested that alterations in the calcium concentration level significantly influence cell stiffness and adhesion, as well as cell migration. In the present study, calcium concentrations were chosen according to the calcium concentration in the synovial fluid of patients with knee osteoarthritis. The results provide solid evidence that the abnormal calcium concentration found in pathological OA conditions could play an important role in the progression of OA. Therefore, targeting calcium concentration might be a promising way of treating OA.

## 4. Materials and Methods

### 4.1. Cell Isolation and Cell Culture

Primary articular chondrocytes were obtained from the knee cartilage of OA patients (*n* = 3, male 62Y, female 64Y, and female 65Y) after total knee arthroplasty with IRB approval. Cartilage tissues used for chondrocytes isolation were chosen from those that were intact according to Osteoarthritis Research Society International (OARSI) criteria. Cartilage tissues were washed three times with PBS and cut into 1 mm^3^ pieces. The collected tissues were digested for 30 min in 0.25% trypsin (Beyotime, Shanghai, China) at 37 °C, and the solution was replaced by 0.15% type II collagenase (Gibco, Carlsbad, CA, USA) in high-glucose DMEM (Gibco, Carlsbad, CA, USA) for 4 h at 37 °C. After digestion, the solution was centrifuged at 1000 rpm for 5 min; then, the supernatant was removed and high-glucose DMEM was supplemented with 10% FBS (Gibco, Carlsbad, CA, USA), before 1% penicillin-streptomycin was added. The obtained chondrocytes were cultured at 37 °C with 5% CO_2_. Four days later, the media was replaced and then changed three times per week. All cells were used within two passages.

### 4.2. AFM Nanoindentation

After 3 days of incubation, the elastic modulus of chondrocytes was measured by an AFM (JPK Instruments, Berlin, Germany). Three separate dish samples were used for each group of mechanical measurements. The silicon-nitride AFM tip (NT-MDT, Zelenograd, Russia) was milled into a flat-end cylinder shape with 2.2 μm diameter, fabricated using a focus ion beam (FEI company, Hillsboro, OR, USA). The spring constant of the tip was calibrated using the integrated thermal noise module of AFM, and was 0.048 N/m. The indentation was performed by controlling the piezoelectric sample stage movement. First, the stage was extended to 8 μm with a speed of 1 μm/s, and then held for 20 s, followed by a retraction of 8μm at a speed of 1 μm/s. Since a rather soft AFM tip was used, the actual indentation depth of the tip into the sample was around 0.5 μm. The cell height was around 6 µm, measured by a confocal laser scanning microscope before the experiment (Appendix A). The indentation location was at the center of the cell. As the cells are known to be highly viscoelastic, to minimize the viscous effects during measurements, in this study, we employed our previously established rate-jump model for mechanical data analysis [49,50]. The elastic modulus was calculated by the Equation [50]:(1)$$\frac{\Delta \dot{\delta }}{\Delta \dot{D}}=A\left(1+\frac{k}{2aE_{r}}\right)$$
where $\Delta \dot{\delta }$ is the net of piezoelectric sample stage speed just before and after the retraction and $\Delta \dot{D}$ is the corresponding rate change in the recorded photodiode signal *D*. *a* is the tip-sample contact radius, and is equal to the tip radius of the flat punch tip used.

### 4.3. Cell Adhesion Measurement

Chondrocytes ECM was extracted from the disposed knee cartilage tissue after total joint replacement. The minced cartilage samples were suspended in distilled water and homogenized with a homogenizer, before the tissue suspension was centrifuged and the supernatant was collected [51]. To enhance its hydrophilic qualities, the AFM tip was treated with plasma machine (Tonson High-Tech Automation Equipment Co., Ltd., Shenzhen, China) under an O_2_ atmosphere for 1 min. The treated AFM tip was immediately submerged in the supernatant for 15 min to obtain the ECM-coated AFM tip. To obtain the adhesion behavior of chondrocytes with ECM, the ECM-coated AFM tip was brought into contact with chondrocytes for 20 s, before being retracted at a speed of 1 μm/s. The AFM height and pulling forces were recorded. Three separate samples were used for each group of mechanical measurements. The maximum unbinding forces between ECM and the cell, and the detachment work, were measured by JPKSPM Data Processing (version: smp-5.0.135, JPK Instruments, Berlin, Germany). An uncoated tip was set as a control group. In the integrin-inhibiting experiment, 10 nM CWHM-12 (MedChemExpress, NJ, USA) was added to the culture media 30 min before measurement. The AFM operation procedures were the same as above.

### 4.4. Confocal Laser Scanning Microscope Observation

After 3-days incubation with culture media containing different [Ca^2+^], chondrocytes were washed three times with PBS and fixed by 4% paraformaldehyde for 20 min. Then, chondrocytes were washed three times with PBS and permeabilized by 0.5% Triton X-100 (Sigma, Saint Louis, MO, USA) for 10 min. Next, chondrocytes were washed three times and incubated with 200 μM phalloidin-TRITC (Sigma, Saint Louis, MO, USA) for 40 min. After being washed with PBS, samples were incubated with Antifade Solution including DAPI (Beyotime, Shanghai, China) for 1 h.

For myosin and integrin immunofluorescence staining, after washing three times with PBS and fixed by 4% paraformaldehyde (Beyotime, Shanghai, China) for 20 min, samples were incubated with 10% goat serum (Beyotime, Shanghai, China) for 30 min. Then, chondrocytes were incubated with primary antibodies, none-muscle myosin ⅡB (Boster, Wuhan, China, 1:400 dilution) and integrin beta 1 (Bioss, Beijing, China, 1:400 dilution) in 1% bovine serum albumin (BSA, Beyotime, Shanghai, China) overnight for 4 °C. Samples were then incubated with the secondary antibody (Cy3 labeled, Sigma, Saint Louis, MO, USA) for two hours at 37 °C. All images were captured under the same conditions. The mean intensity of the F-actin was measured by ImageJ software (version: version: 1.53K; National Institutes of Health, Bethesda, MD, USA).

### 4.5. Quantitative Real-Time PCR

After 3 days of incubation with culture media containing different [Ca^2+^]_,_ chondrocytes mRNA was collected and purified using RNAiso Plus (TaKaRa, Dalian, China). The mRNA was dissolved in DEPC-treated water and quantified using Nanodrop (ThermoFisher, Carlsbad, CA, USA). Then, mRNA was transcribed to cDNA using the cDNA Synthesis Kit (Beyotime, Shanghai, China). The gene expression levels were detected with an SYBR Green real-time PCR Master Mix kit (Beyotime, Shanghai, China) and performed on an Applied Biosystems, StepOnePlus Instrument (Thermofisher, Carlsbad, CA, USA). The gene expression levels were normalized by *β-actin*; 2^−∆∆ct^ was calculated by referring to the 1.25 mM calcium ion group. All the primers were shown in Appendix A. 

### 4.6. Wound Healing Assay

The migration ability of chondrocytes at different [Ca^2+^] levels was investigated by a wound-healing assay. The ECM solution was employed to clean the used six-well plate for 10 min. Chondrocytes were seeded on the dried six-well plate with culture media containing different [Ca^2+^]. When cells reached full confluence, the cell monolayer was scratched with a 100 μL pipette tip. Images were captured right after scratching (0 h) and 48 h later using a Leica DMi8 microscope. The images were analyzed with ImageJ software. Six areas were selected to measure the width of the scratched area, and the migration distance was obtained by the decrease in the wound width for 48 h.

### 4.7. Western Blot

After 3 days incubation with a culture medium containing [Ca^2+^], chondrocyte proteins were collected with RIPA buffer. The total protein concentrations were measured by BCA Protein Assay Kit (Beyotime, Shanghai, China). The primary antibodies are shown below: anti-integrin beta 1 antibody (Abcam, MA, USA), anti-beta Actin antibody (Abcam, Boston, MA, USA), anti-non-muscle Myosin IIB/MYH10 antibody (Boster, Wuhan, China).

### 4.8. Statistical Analysis

All the experiments were carried out independently and in triplicate, and all data were expressed as mean ± SD. Statistical analysis was carried out using *ANOVA* and *Tukey’s* post hoc in Graphpad (version: 8.0, GraphPad Software Inc., San Diego, CA, USA); if a *p*-value was less than 0.05, it was considered statistically significant. The results of the wound-healing assay and fluorescence staining were quantified by ImageJ software.

## 5. Conclusions

In the current study, we investigated the effect of the cell elastic modulus and cell–ECM adhesion under the influence of the extracellular calcium ions in the human primary chondrocytes. We found that chondrocyte migration in vitro was regulated by extracellular calcium ions by modulating cell stiffness and cell–ECM adhesion. With the pathological increase in extracellular calcium ions, the elastic modulus and cell–ECM adhesion forces of the chondrocytes initially increase, followed by a decrease. Under the influence of the mechanical property of the chondrocytes, chondrocyte migration also shows the same pattern, and is affected by the extracellular calcium ions. The expression of *myosin II* and *integrinβ1* may be responsible for the aforementioned changes. Lastly, we believe that the findings in the study will help provide a better understanding of the role of calcium in chondrocyte function and possible OA progression.

## Figures and Tables

**Figure 1 ijms-22-10034-f001:**
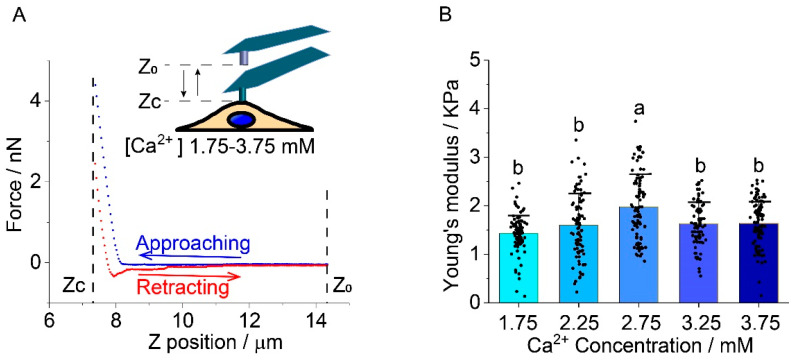
The effects of extracellular Ca^2+^ concentration on human chondrocyte’s stiffness. (**A**) A representative AFM indentation curve measured for a chondrocyte cultured at 1.75 mM Ca^2+^ concentration. Z0 is the initial position of the AFM tip, Zc is the constant holding position of the tip; (**B**) As the [Ca^2+^] increases, from the normal physiological level (1.75 mM), the Young’s modulus of chondrocytes increases by 37% at 2.75 mM and declines at a higher Ca^2+^ concentration. N = 90 cells/group are measured and analyzed. Group difference was detected with *ANOVA* and Tukey’s post hoc tests in GraphPad Prim 8. Letters (a, b) above the bars correspond to distributions that are (letters different) or are not significantly different (letter shared) at the *p* < 0.05 level.

**Figure 2 ijms-22-10034-f002:**
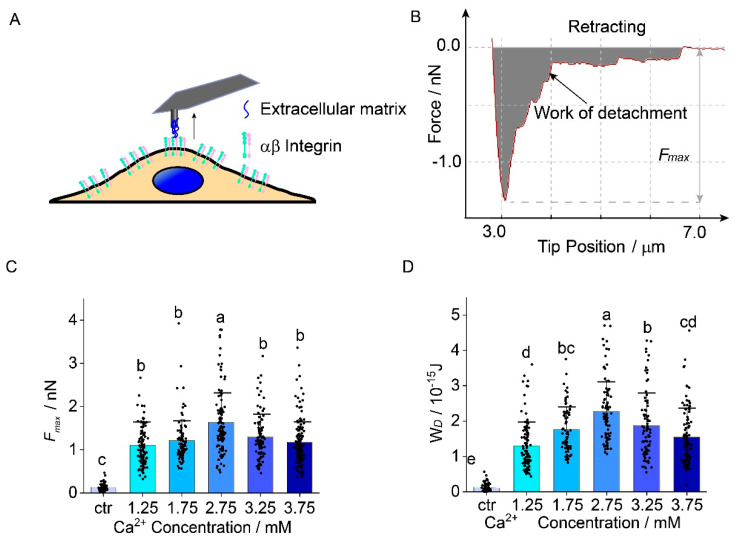
The effects of extracellular Ca^2+^ concentration on adhesion between human chondrocytes and extracellular matrix. (**A**) Schematic of the measurement for the adhesion between chondrocytes and extracellular matrix on AFM tip; (**B**) Example of a retraction force-distance curve obtained by cell to extracellular matrix adhesion measurement. *F_max_* is the difference between the peak force value and the baseline of complete detachment. Work of detachment (The gray region) represents the energy during the detachment process; (**C**) As Ca^2+^ concentration increases, forces between chondrocytes and extracellular matrix increase by 48.0% and reach the peak at 2.75 mM Ca^2+^ concentration then decrease by 26% Ca^2+^ concentration from 2.75 mM to 3.75 mM; (**D**) The energy during the detachment process increases by 72% from Ca^2+^ concentration 1.75 mM to 2.75 mM, decreases by 31.3% from Ca^2+^ concentration 2.75 mM to 3.75 mM. 90 cells for each group are measured and analyzed. *ANOVA* and Tukey’s post hoc is performed for data analysis. Letters (a, b, c, and d) above the bars correspond to distributions that are (letters different) or are not significantly different (letter shared) at the *p* < 0.05 level.

**Figure 3 ijms-22-10034-f003:**
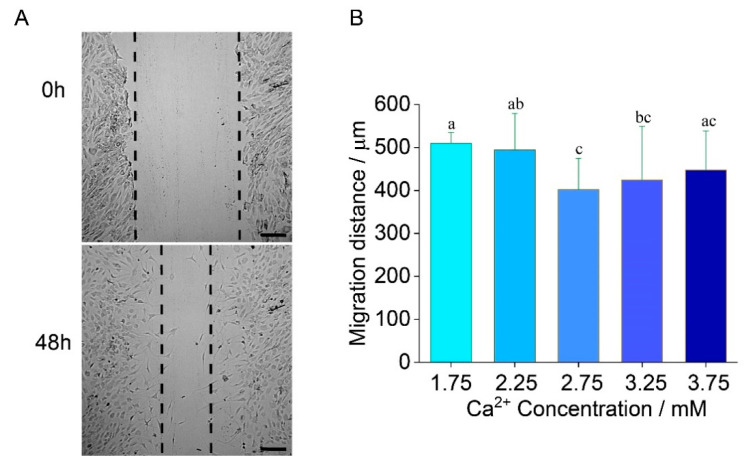
Human chondrocytes migrate at different Ca^2+^ concentrations in 48 h. (**A**) A representative scratch assay at 1.75 mM Ca^2+^ concentration. Scale bar 250 μm; (**B**) Cell migration distance decreases by 21% from Ca^2+^ concentration 1.75 mM to 2.75 mM. 6 pictures for each group are measured and analyzed. *ANOVA* and Tukey’s post-hoc is performed for data analysis. Letters (a, b, c) above the bars correspond to distributions that are (letters different) or are not significantly different (letter shared) at the *p* < 0.05 level.

**Figure 4 ijms-22-10034-f004:**
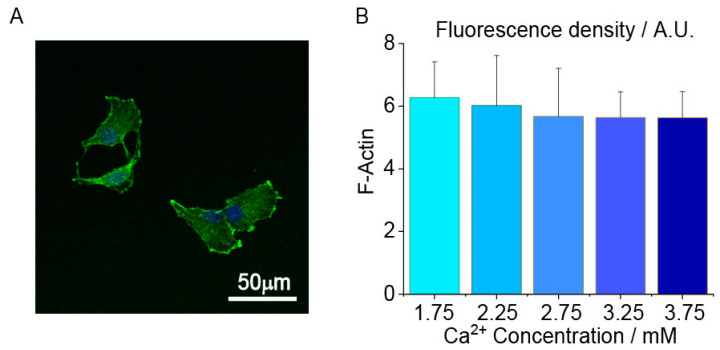
The effects of extracellular Ca^2+^ concentration on human chondrocytes actin filament fluorescence density. (**A**) A representative chondrocytes actin fluorescence image taken at 1.75 mM Ca^2+^ concentration. Scale bar 50 μm; (**B**) No significant difference in the average fluorescence density of each group. 60 cells are measured and analyzed in each group. *ANOVA* and Tukey’s post hoc is performed for data analysis. Cell mechanics and cell–ECM adhesion are regulated by myosin and integrin expression at different [Ca^2+^]. Myosin, as one of the components of the cytoskeleton, can bond with actin and will influence the stiffness of cells (Figure 5A). During the migration, cells need to dissociate adhesions and form a new integrin–ECM adhesion (Figure 5B). The expression of *myosin* increased to the peak value from 1.75 mM to 2.75 mM, and then decreased from 2.75 mM to 3.75 mM [Ca^2+^] (Figure 5C). The expression of *integrinβ1* increased in the lower [Ca^2+^] (e.g., 1.75 mM and 2.75 mM) and decreased in the higher [Ca^2+^] (e.g., 2.75 mM and 3.75 mM). (Figure 5D). The expression of *integrinβ3* showed no difference between each group (Appendix A).

**Figure 5 ijms-22-10034-f005:**
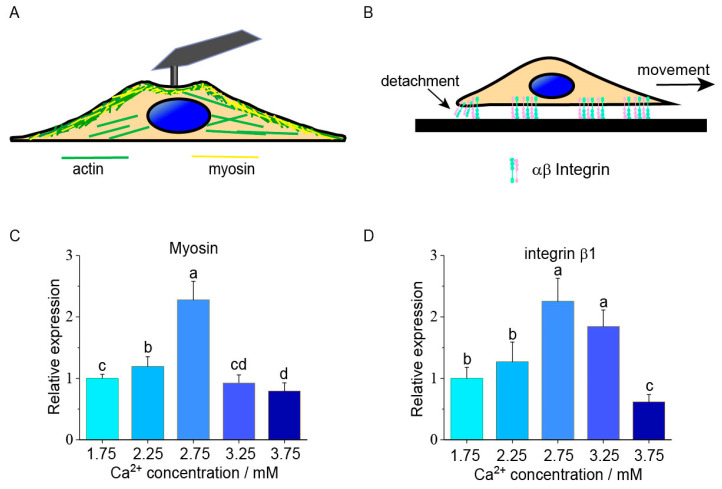
Expression of cell movement-related genes at different Ca^2+^ concentrations. (**A**) Schematic presentation of the *actin* and *myosin* as the supporting structure in human chondrocytes; (**B**) *Integrin* as an adhesion molecule for cell and extracellular matrix; (**C**,**D**) *myosin* and *integrinβ1* expression levels show an increasing and decreasing trend, which reach a peak at 2.75 mM Ca^2+^ concentration. *ANOVA* and Tukey’s post-hoc is performed for data analysis. Letters (a, b, c, and d) above the bars correspond to distributions that are (letters different) or are not significantly different (letter shared) at the *p* < 0.05 level.

**Figure 6 ijms-22-10034-f006:**
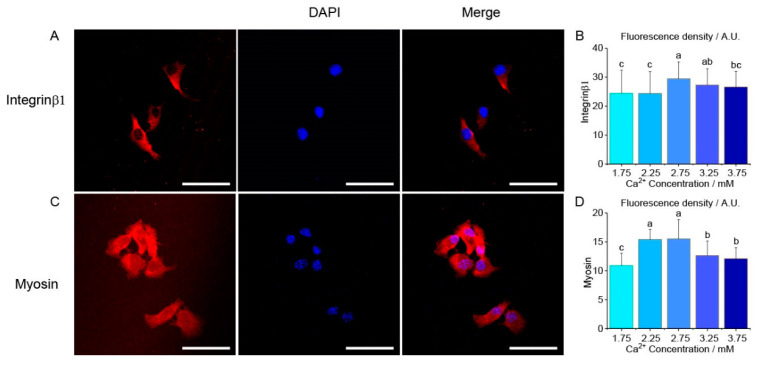
The effects of extracellular Ca^2+^ concentration on human chondrocytes myosin and integrinβ1 fluorescence density. (**A**) A representative integrinβ1 fluorescence image taken at 1.75 mM Ca^2+^ concentration. Scale bar 50 μm; (**B**) As Ca^2+^ concentration increases, fluorescence density of integrinβ1 increased by 20% from 1.75 mM to 2.75 mM, and then decreased from the 2.75 mM group to 3.75 mM. (**C**) A representative myosin fluorescence image taken at 1.75 mM Ca^2+^ concentration. Scale bar 50 μm; (**D**) As Ca^2+^ concentration increases, fluorescence density of myosin increased by 50% from 1.75 mM to 2.75 mM and then decreased by 20% from 2.75 mM to 3.75 mM. 60 cells are measured and analyzed in each group. Letters (a, b, c) above the bars in (**B**,**D**) correspond to distributions that are (letters different) or are not significantly different (letter shared) at the *p* < 0.01 level.

**Figure 7 ijms-22-10034-f007:**
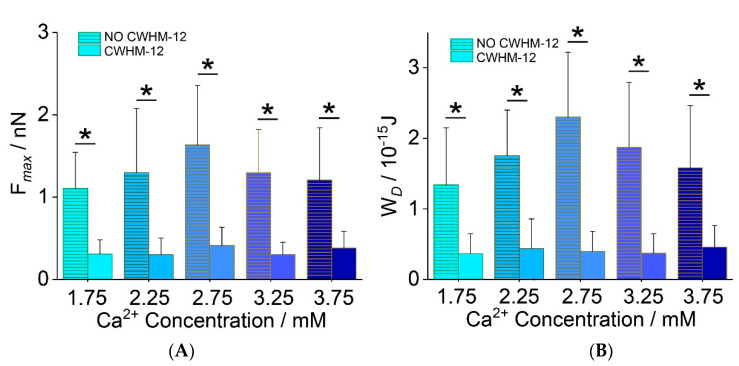
The adhesion between human chondrocytes and ECM after CWHM-12 treatment at different extracellular Ca^2+^ concentrations. The value of *F_max_* in five groups was between 300 ± 200 pN (2.25 mM) and 410 ± 217 pN (2.75 mM) (**A**). The value of *W_d_* in five groups was between 3.6 ± 2.8 × 10^−16^ J (1.75 mM) and 4.6 ± 3.1 × 10^−16^ J (3.75 mM) (**B**). Compared with non-treatment groups, *F_max_* in CWHM-12 groups dramatically decreased. *: *p* < 0.05, 60 cells were measured and analyzed in each group.

## Data Availability

The data presented in this study are available on request from the corresponding author.

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
