# Peer review of "Extracellular Calcium Ion Concentration Regulates Chondrocyte Elastic Modulus and Adhesion Behavior"

_ijms, 2021, doi:10.3390/ijms221810034_

Round 1
Reviewer 1 Report
This article shows the effects of calcium ion in circumstance on the migration via cell stiffness and adhesion factors of chondrocytes. The authors assayed with the unique methods and the findings can be useful to clinical diagnose.
However, there are several points to be revised as described below.
Major points
The activity and expression level of ECM degradation enzymes (e.g., MMPs) should be assayed. Because they are noteworthy as same as adhesion factors in migration of chondrocyte and MMPs activity is influenced by calcium ion.
Why did authors select CWHM-12, a inhibitor of integrin av? Though they always focused to integrin b1 in this article.
In Figure7, both data with and without CWHM-12 should be shown. It is impossible to compare the value to the value from another experiment.
Measure intracellular calcium concentrations in all conditions.
Minor points
The pictures in Figure3A are not enough clear to recognize cells. Authors should change them to higher resolution images or stain cells with toluidine blue or something.
In FigureS3, the density of each blot should be measured.
FigureS2 does not indicate essential data: the informations of each color, apical side, basal side.
Generally, unit in figure axis indicated in brackets can be better than slash.
To confirm the cell shape, show the phase contrast image of cells in all dose of calcium ion.
Author Response
Reviewer 1:
This article shows the effects of calcium ion in circumstance on the migration via cell stiffness and adhesion factors of chondrocytes. The authors assayed with the unique methods and the findings can be useful to clinical diagnose.
However, there are several points to be revised as described below.
Major points
The activity and expression level of ECM degradation enzymes (e.g., MMPs) should be assayed. Because they are noteworthy as same as adhesion factors in migration of chondrocyte and MMPs activity is influenced by calcium ion.
Response: Thanks for the reviewer’s advice on detecting the expression level of MMPs. It is rational and valuable as the cell migration process involves both degradation and adhesion to ECM. However, degradation and adhesion are two disparate cell behaviors, discussing both of them in one article will make the article be unfocused and ambiguous. In this article, we mainly focus on the mechanics-related factors such as cell stiffness and cell-ECM adhesion. The reviewer’s advice is enlightening, we will consider it in the following work.
Why did authors select CWHM-12, a inhibitor of integrin av? Though they always focused to integrin b1 in this article.
Response: thanks for your careful review. The CWHM-12 acts on integrin av, but also have a function on integrin beta 1 with higher concentration as we used in this article. Another objective reason is that only this drug is available during the experiment period. The results also show that the inhibition effect is obvious.
In Figure7, both data with and without CWHM-12 should be shown. It is impossible to compare the value to the value from another experiment.
Response: thanks for your advice. We have revised figure 7 as shown below.
Measure intracellular calcium concentrations in all conditions.
Response: Thanks for your advice. It is a very good idea to measure the intracellular calcium concentration to verify the potent signal transduction. We admit the limitation in this article that the accurate mechanism that how extracellular calcium ion transduces to intracellular and how calcium ion influence integrin and myosin are speculated from references but not verified by our experiments.
Minor points
The pictures in Figure3A are not enough clear to recognize cells. Authors should change them to higher resolution images or stain cells with toluidine blue or something.
Response: thanks for your advice. The picture has been revised and shown below.
In FigureS3, the density of each blot should be measured.
Response: thanks for your advice. The density of each blot is shown below.
FigureS2 does not indicate essential data: the informations of each color, apical side, basal side.
Response: thanks for your advice. The detailed information has been supplemented in figure S2.
Figure S1. The height of chondrocytes was about 6mm measured by confocal laser scanning microscope. Blue: DAPI; green: phalloidin.
To confirm the cell shape, show the phase contrast image of cells in all doses of calcium ion.
Response: Thanks for your advice. We show you the phalloidin staining images of all doses of calcium concentration No obvious difference is found among each group. Images from left to right correspond to 1.75 mM to 3.75mM. scale bar: 100 mm.

Reviewer 2 Report
This is an interesting study on the role of extracellular calcium on chondrocyte elastic modulus, migration and adhesion. The manuscript is well written and may enhance the Journal's audience. However, some changes should be made:
- English language editing and polishing is advised (e.g. the article in the title "the" may be removed; line 29, "increasing" should be changed to "increased"; line 61, "some certain" is a tautology, please remove "some"; line 195, please remove the possessive here; line 197, please remove "to" following "helps", please change "abnormally" to "abnormal"; etc.)
- A space between the text and reference in brackets should be inserted wherever applicable. The same should be applied to digits and corresponding units of measurement (e.g. line 89, 1.75mM).
- The ionic charge (2+) should be written as superscript (i.e. Ca2+).
- Lines 35-36: based on the latest findings, it is commonly accepted that OA is a disease involving the whole joint and not only articular cartilage. Authors should correct this sentence accordingly.
- Line 42: the abbreviated form [Ca2+]o is introduced here for the first time. The extended form should be mentioned here with the abbreviated form in brackets. I suggest that authors remove the small "o" after [Ca2+] for better clarity.
- Lines 49, 128 and elsewhere: the word "wound" does not seem appropriate here. Please change it to "damage".
- The Results section should be reorganized in subsections (e.g. 2.1. Chondrocyte stiffness; 2.2. Chondrocyte adhesion to extracellular matrix; 2.3. Chondrocyte migration).
- The discussion should start with a paragraph resuming the available literature on the role of extracellular calcium in physiological and osteoarthritic chondrocytes.
- References should be formatted according to Instructions for Authors.
Author Response
Reviewer 2
This is an interesting study on the role of extracellular calcium on chondrocyte elastic modulus, migration and adhesion. The manuscript is well written and may enhance the Journal's audience. However, some changes should be made:
- English language editing and polishing is advised (e.g. the article in the title "the" may be removed; line 29, "increasing" should be changed to "increased"; line 61, "some certain" is a tautology, please remove "some"; line 195, please remove the possessive here; line 197, please remove "to" following "helps", please change "abnormally" to "abnormal"; etc.)
Response: Thanks for your patient review. We have revised these mistakes.
- A space between the text and reference in brackets should be inserted wherever applicable. The same should be applied to digits and corresponding units of measurement (e.g. line 89, 1.75mM).
Response: Thanks for your review, we have added space to these sites..
- The ionic charge (2+) should be written as superscript (i.e. Ca2+).
Response: Thank you. We have revised them.
- Lines 35-36: based on the latest findings, it is commonly accepted that OA is a disease involving the whole joint and not only articular cartilage. Authors should correct this sentence accordingly.
Response: Thanks for your advice. We have revised the text. “Osteoarthritis (OA) is a common joint disorder and a leading cause of physical disability, one of the characters is articular cartilage degeneration.”
- Line 42: the abbreviated form [Ca2+]o is introduced here for the first time. The extended form should be mentioned here with the abbreviated form in brackets. I suggest that authors remove the small "o" after [Ca2+] for better clarity.
Response: Thanks for your advice. We have deleted the “o”.
- Lines 49, 128 and elsewhere: the word "wound" does not seem appropriate here. Please change it to "damage".
Response: Thanks for your advice. We have changed it to “damage”.
- The Results section should be reorganized in subsections (e.g. 2.1. Chondrocyte stiffness; 2.2. Chondrocyte adhesion to extracellular matrix; 2.3. Chondrocyte migration).
Response: Thanks for your advice. We have added the subtitle in Results Section. “2.1 Chondrocyte stiffness 2.2. Chondrocyte adhesion to ECM 2.3. Chondrocyte migration 2.4 Chondrocyte cytoskeleton and adhesion-related proteins expression 2.5 Chondrocyte adhesion to ECM after integrin inhibition”.
- The discussion should start with a paragraph resuming the available literature on the role of extracellular calcium in physiological and osteoarthritic chondrocytes.
Response: Thanks for your advice. we have added a paragraph in the discussion.
“Over the last years, high concentration of calcium in the joint fluid of OA patients has emerged as an important hallmark in the pathogenesis of OA as well as a critical simulation to cell. Despite its physiological importance, the relationship among high [Ca2+] level and chondrocyte mechanics and chondrocyte migration are still be ambiguous. In this investigation, we measured chondrocyte stiffness and chondrocyte-ECM adhesion and explained their relationship with cell migration. Furthermore, we demonstrated the critical role of myosin and integrin in chondrocyte mechanics and migration. Our work makes an important advance in understanding the role of [Ca2+] in OA pathogenesis and also helps reveal the correlation of cell mechanics and cell migration.”
- References should be formatted according to Instructions for Authors.
Response: Thanks for your advice. We have revised the format of the reference.

Reviewer 3 Report
Please find hereby my review of the article untitled “Extracellular Calcium ion Concentration Regulates the Chondrocyte Elastic Modulus and Adhesion Behavior” by Shen Xingyu et al. in International journal of molecular sciences.
The submitted manuscript has a very interesting topic and a real scientific interest.
Authors measured the elastic modulus of chondrocytes and cell-ECM adhesion force at different [Ca2+] using human primary chondrocytes, using atomic force microscopy (AFM). They quantified the migration abilities of chondrocytes and analyzed the cytoskeleton structure. Moreover, several cytoskeletons related genes and proteins expressions were investigated to explore the possible reasons for the observed changes in cell mechanical properties with different [Ca2+]. Taken together, the results showed that [Ca2+] could regulate chondrocytes stiffness and cell-ECM adhesion, and consequently, influence cell migration, which is critical for cartilage repair.
The manuscript needs to be edited carefully (typographical errors…).
Authors must show viability tests as MTT LDH Proliferation assay
Do you have check autophagy process?
Authors must show gene expression of other ECM or attachments protein, others integrin
Do have you check expression of ciliary component?
What the Effect of high calcium on ERK P38 JNK activation?
Do you have observed Crystal formation with high Concentration of calcium (Calcium phosphate, BCP or others)?
What is the calcium impact on the Mechanical transduction (pression/compression) as these elements was very important in cartilage biology???
You use High glucose concentration in the medium (4,5 g/l), you need to test with 1 g/l (normal concentration of glucose in the body) as numerous studies demonstrate the impact of high glucose on chondrocytes response.
I didn’t show any modification of myosin and integrin b1 expression in S3 supplementary file as the level of b-actin was also increase with the 2,75 mM compared to the control. Need to confirmed these experiments and to make density analyses and show the ratio with b-actin.
3 days of treatment with calcium was very long? What results for shorter times? A few hours? 24 h?
Author Response
Reviewer 3:
Please find hereby my review of the article untitled “Extracellular Calcium ion Concentration Regulates the Chondrocyte Elastic Modulus and Adhesion Behavior” by Shen Xingyu et al. in International journal of molecular sciences.
The submitted manuscript has a very interesting topic and a real scientific interest.
Authors measured the elastic modulus of chondrocytes and cell-ECM adhesion force at different [Ca2+] using human primary chondrocytes, using atomic force microscopy (AFM). They quantified the migration abilities of chondrocytes and analyzed the cytoskeleton structure. Moreover, several cytoskeletons related genes and proteins expressions were investigated to explore the possible reasons for the observed changes in cell mechanical properties with different [Ca2+]. Taken together, the results showed that [Ca2+] could regulate chondrocytes stiffness and cell-ECM adhesion, and consequently, influence cell migration, which is critical for cartilage repair.
The manuscript needs to be edited carefully (typographical errors…).
Response: Thanks for your advice. We have revised the manuscript.
Authors must show viability tests as MTT LDH Proliferation assay
Response: Thanks for your advice. We have supplemented the MTT assay in the supplementary results. The assay measured the proliferation rate of chondrocyte from day1 to day 3, the results show no difference between each group at day 3.
Do you have check autophagy process?
Response: Thanks for your question. In this article, we do not check the autophagy process
Authors must show gene expression of other ECM or attachments protein, others integrin
Response: Thanks for your advice. The principal components of articular cartilage are collagen II and Acan. In supplementary S2, we show the expression of ECM proteins such as collagen I, collagen Ⅱ and Acan. The results show no difference between each group. Even though there are some other ECM proteins in cartilage, they are limited and too complicated to detect all of them. Meanwhile, we agree that ECM proteins play a critical role in cell migration. However, in this article, we mainly focus on the variance of calcium concentration but not the change of ECM. We appreciate the reviewer’s advice on detecting the ECM expression with different calcium concentrations, we should consider it in future work. As for integrin expression, we also detect the expression of integrin beta 3 as shown in Supplementary S2. Because integrin beta 1 and 3 are mainly adhesive molecules in mammal cells, we focus on these two types of integrin. The integrin beta 3 does not show the difference in each group, so we detect the expression of integrin beta 1 with WB, IF and qPCR. Thanks for your advice, hope our explanation makes sense.
Do have you check expression of ciliary component?
Response: Thanks for your question. In this article, we do not check the ciliary component.
What the Effect of high calcium on ERK P38 JNK activation?
Response: Thanks for your question. Many articles reported that calcium can regulate ERK signal[1], but I do not find related articles in human chondrocytes. Some articles reported that octacalcium phosphate crystal induced p38 and JNK MAPK phosphorylation in calves chondrocyte[2]. It is very intriguing to investigate these signal transductions in human chondrocytes. Thanks for your enlightenment.
Do you have observed Crystal formation with high Concentration of calcium (Calcium phosphate, BCP or others)?
Response: Thanks for your question. Many articles reported that crystal formation in osteoarthritis. In our research, these in vitro experiments do not show crystal in cell culture medium.
What is the calcium impact on the Mechanical transduction (pression/compression) as these elements was very important in cartilage biology???
Response: Thanks for this nice question. Many articles reported the calcium signaling activated by mechanical cues such as compression[3, 4], but seldom noticed the high calcium concentration influences the mechanotransduction. It probably has an influence on mechanotransduction as calcium signal is closely related to mechanotransduction molecular such as cadherin and TRPV4 channel. Thanks for your inspiration.
You use High glucose concentration in the medium (4,5 g/l), you need to test with 1 g/l (normal concentration of glucose in the body) as numerous studies demonstrate the impact of high glucose on chondrocytes response.
Response: Thanks for your question. We agree with the reviewer’s opinion that it is better to simulate the physiological condition, but in vitro cell culture condition is not completely consistent with the one in vivo. The reason we choose high glucose culture medium is that chondrocytes grow well in this culture medium, cell proliferate normally.
I didn’t show any modification of myosin and integrin b1 expression in S3 supplementary file as the level of b-actin was also increase with the 2,75 mM compared to the control. Need to confirm these experiments and to make density analyses and show the ratio with b-actin.
Response: Thanks for your advice. We have revised figure S3 and added the density analyses in the figure.
3 days of treatment with calcium was very long? What results for shorter times? A few hours? 24 h?
Response: Thanks for your question. It is interesting to test the cell response with short time calcium treatment. Concerned about the clinical condition, chondrocyte in OA cartilage chronically stay in the high calcium concentration, that is the reason we choose the long-time treatment of calcium. Hope this answer makes sense.

Reviewer 4 Report
In this study, biophysical and cell biological effects of calcium ion concentration on osteoarthritic chondrocytes. The stiffness, maximum unbinding force and work of detachment as measured by atomic force microscopy all reached a peak at 2.75 mM of calcium ion concentration, whereas cell migration showed an opposite trend. Actin filament showed no appreciable difference in the presence of different calcium ion concentration, but myosin and integrin beta 1 production was increased up to 2.75 mM with subsequent decrease along with the concentration. Experiments with an integrin inhibitor confirmed the contribution of integrins to the regulation of maximum unbinding force and work of detachment by calcium ion concentration.
Overall, the presented data are clear and novel, and scientifically significant discussion is made based on these data. However, several points are left to be addressed.
Specific points
1) Nothing is visible in the images presented in Figure 3A, except for lines and scale bars. Please provide another set of images with better quality.
2) The authors should explain what "a", "b" and "c" in Figure 3B stands for more in detail.
3) Probably, present Figure 7 is not in the best fashion in showing the effect of the integrin inhibitor on maximum unbinding force and work of detachment under a variety of calcium ion concentrations. The authors should show the data in such a manner that readers could tell the significant effects of the inhibitor without reading the corresponding text or going back to Figure 2.
4) Strictly speaking, the data in Figure 2 may not directly represent "adhesion between human chondrocytes and extracellular matrix", and thus this phrase may be reconsidered.
5) According to the 2nd paragraph of Discussion, a previous study indicated that at 1 - 10 mM calcium ion concentration, it bound to ADMIDAS. However, the authors insist that between 1.75 mM and 2.75 mM calcium (both 1 - 10 mM) ion concentration, it bound to LIMBS and MIDAS in this study. These statements are contradictory and thus should be reconsidered.
Minor points
1) Please explain what CWHM-12 is at the first use.
2) Although English is overall written well, minor errors and mistakes are still found. For example, the 2nd sentence in the 2nd paragraph in Discussion should read "Cell-ECM adhesion is mainly mediated by...", and the third sentence should be "As chondrocyte ECM mainly...". Careful English editing is recommended.
Author Response
Reviewer 4:
In this study, biophysical and cell biological effects of calcium ion concentration on osteoarthritic chondrocytes. The stiffness, maximum unbinding force and work of detachment as measured by atomic force microscopy all reached a peak at 2.75 mM of calcium ion concentration, whereas cell migration showed an opposite trend. Actin filament showed no appreciable difference in the presence of different calcium ion concentration, but myosin and integrin beta 1 production was increased up to 2.75 mM with subsequent decrease along with the concentration. Experiments with an integrin inhibitor confirmed the contribution of integrins to the regulation of maximum unbinding force and work of detachment by calcium ion concentration.
Overall, the presented data are clear and novel, and scientifically significant discussion is made based on these data. However, several points are left to be addressed.
Specific points
1) Nothing is visible in the images presented in Figure 3A, except for lines and scale bars. Please provide another set of images with better quality.
Response: Thanks for your suggestion, we have revised figure 3A and shown it below.
2) The authors should explain what "a", "b" and "c" in Figure 3B stands for more in detail.
Response: Thanks for your advice. columns with different letters mean the significant difference between them; on the contrary, columns which contain the same letters have no difference.
3) Probably, present Figure 7 is not in the best fashion in showing the effect of the integrin inhibitor on maximum unbinding force and work of detachment under a variety of calcium ion concentrations. The authors should show the data in such a manner that readers could tell the significant effects of the inhibitor without reading the corresponding text or going back to Figure 2.
Response: Thanks for your suggestion. We have revised Figure 7 as shown below.
4) Strictly speaking, the data in Figure 2 may not directly represent "adhesion between human chondrocytes and extracellular matrix", and thus this phrase may be reconsidered.
Response: Thanks for your advice. How about change to “adhesion between human chondrocytes and extracted extracellular matrix”?
5) According to the 2nd paragraph of Discussion, a previous study indicated that at 1 - 10 mM calcium ion concentration, it bound to ADMIDAS. However, the authors insist that between 1.75 mM and 2.75 mM calcium (both 1 - 10 mM) ion concentration, it bound to LIMBS and MIDAS in this study. These statements are contradictory and thus should be reconsidered.
Response: Thanks for your advice. It is true that our statement has some problems. We have revised the text according to yout advice.
“Our results demonstrate that cell-ECM adhesion can be regulated by extracellular cal-cium concentration through integrinβ1. Calcium ions may regulate integrin from the following aspects. Firstly, integrin-mediated adhesion can be regulated by calcium signals from inside and outside of the cells. Integrin mediates calcium signaling from the extracellular domain (‘outside-in’ signaling), and then activates intracellular signal pathways. Intracellular feedback signals subsequently regulate integrin-mediated ad-hesion (‘inside-out’ signaling)[37, 38]. Secondly, the regulation of Ca2+ to integrin pre-sents a bipolar trend and Ca2+ plays an important role in stabilizing integrin structure to mediate integrin-ligand binding. There are three or four Ca2+ binding sites in integrin, which involve regulating integrin-ligand binding affinity. Calcium ions bind to integ-rin and change the conformation thus influence the integrin-ligand binding affinity[39]. Previous ligand binding assay demonstrated that when calcium ion is in a lower concentration, Ca2+ binds with ligand-bind sites such as LIMBS (lig-and-associated metal-binding site) and MIDAS (metal-ion dependent adhesion site), plays a positive role in integrin-ligand adhesion. When calcium ion increases to a higher concentration level (1-10mM), Ca2+ binds with ligand-bind site such as AD-MIDAS ((adjacent to MIDAS), plays a negative role in integrin-ligand adhesion[40, 41]. In this study, as Ca2+ stay at high concentration, Ca2+ binds with ADMIDAS making more and more integrin keep in low affinity and present smaller detachment force. Taken together, the expression of integrin and integrin affinity change cause the cell-ECM adhesion presents a bipolar trend in the 2.75mM to 3.75mM calcium concentration.”
Minor points
1) Please explain what CWHM-12 is at the first use.
Response: Thanks for your advice. We have added the explanation in the method.
2) Although English is overall written well, minor errors and mistakes are still found. For example, the 2nd sentence in the 2nd paragraph in Discussion should read "Cell-ECM adhesion is mainly mediated by...", and the third sentence should be "As chondrocyte ECM mainly...". Careful English editing is recommended.
Response: Thank you for your advice. We have revised the article and checked it carefully.

Round 2
Reviewer 1 Report
The revised version is better than the previous one. However you should statistically analyzed in Figure 7.
I hope authors will progress this study more.
Author Response
Thanks for your careful review. we have revised figure 7 and showed statistical differences in the picture.
Reviewer 3 Report
The modifications/corrections is now ok for me.
Thank you
Author Response
Thanks for your careful review.